# BUTTERFLYQUANT: ULTRA-LOW-BIT LLM QUANTIZATION THROUGH LEARNABLE ORTHOGONAL BUTTERFLY TRANSFORMS

## ABSTRACT

Large language models require massive memory footprints, severely limiting deployment on consumer hardware. Quantization reduces memory through lower numerical precision, but extreme 2-bit quantization suffers from catastrophic performance loss due to outliers in activations. Rotation-based methods such as QuIP and QuaRot apply orthogonal transforms to eliminate outliers before quantization, using computational invariance: $\mathbf{y} = \mathbf{W}\mathbf{x} = (\mathbf{W}\mathbf{Q}^T)(\mathbf{Q}\mathbf{x})$ for orthogonal $\mathbf{Q}$. However, these methods use fixed transforms–Hadamard matrices achieving optimal worst-case coherence $\mu = 1/\sqrt{n}$–that cannot adapt to specific weight distributions. We identify that different transformer layers exhibit distinct outlier patterns, motivating layer-adaptive rotations rather than one-size-fits-all approaches. In this work, we propose ButterflyQuant, which replaces Hadamard rotations with learnable butterfly transforms parameterized by continuous Givens rotation angles. Unlike Hadamard's discrete $\{+1, -1\}$ entries that are non-differentiable and thus prohibit gradient-based learning, butterfly transforms' continuous parameterization enables smooth optimization while guaranteeing orthogonality by construction. This orthogonal constraint ensures theoretical guarantees in outlier suppression while achieving $O(n \log n)$ computational complexity with only $\frac{n \log n}{2}$ learnable parameters. We further introduce a uniformity regularization on post-transformation activations to promote smoother distributions amenable to quantization. Learning requires only 128 calibration samples and converges in minutes on a single GPU–a negligible one-time cost. For LLaMA-2-7B with 2-bit quantization, ButterflyQuant achieves 15.4 perplexity versus 37.3 for QuIP. The codes are in the Supplementary Materials.

## 1 INTRODUCTION

Large language models (LLMs) have demonstrated remarkable capabilities, but their deployment remains severely constrained by memory requirements. LLaMA-70B requires 140GB in FP16 precision, exceeding the capacity of most GPUs and making consumer deployment infeasible [Touvron et al., 2023, Zhao et al., 2025]. Recent research highlights deployment challenges including memory bandwidth bottlenecks, with inference serving as the dominant cost in production systems [Chen et al., 2024, Yuan et al., 2024]. Quantization—reducing numerical precision to 2-4 bits—offers a direct solution by compressing LLMs 4-8×. However, extreme quantization suffers from catastrophic performance degradation due to outliers in activations that dominate the dynamic range [Dettmers et al., 2022, Wei et al., 2022], a primary obstacle to low-bit compression [Sun et al., 2024].

To mitigate the outlier problem, rotation-based quantization methods have emerged as a robust solution [Ashkboos et al., 2024, Chee et al., 2023]. These methods apply an orthogonal transformation $\mathbf{Q}$ before quantization, leveraging computational invariance: $\mathbf{y} = \mathbf{W}\mathbf{x} = (\mathbf{W}\mathbf{Q}^T)(\mathbf{Q}\mathbf{x})$. The rotation redistributes activations across channels, effectively smoothing out outlier features without altering the layer's output. Prominent methods like QuaRot [Ashkboos et al., 2024] use fixed Hadamard transforms, which achieve optimal worst-case coherence, while QuIP [Chee et al., 2023] employs random orthogonal matrices. Despite their success, these approaches share a critical limitation: they apply a *fixed*, data-agnostic transform with discrete $\{+1, -1\}$ entries that cannot be optimized via gradients to all layers.

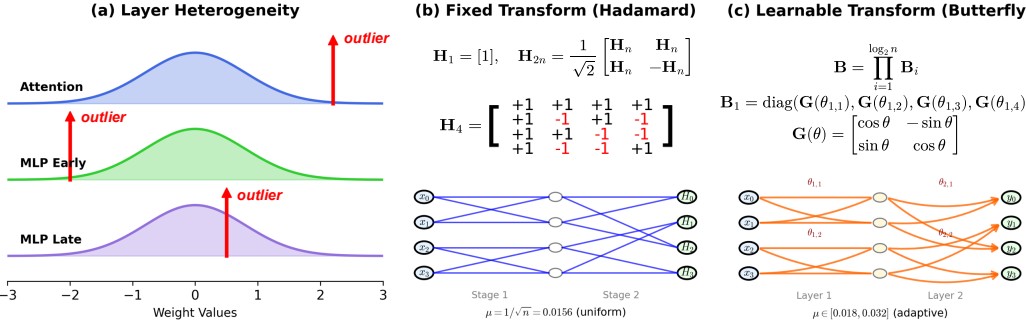

Figure 1: **Layer heterogeneity motivates learnable transforms for LLM quantization. (a)** Different transformer layers exhibit distinct outlier distributions: attention (positive tails), early MLP (negative regions), late MLP (boundaries). **(b)** Hadamard transforms with discrete $\{+1, -1\}$ entries apply fixed rotations through recursive decomposition $\mathbf{H}_{2n} = \frac{1}{\sqrt{2}}[\mathbf{H}_n, \mathbf{H}_n; \mathbf{H}_n, -\mathbf{H}_n]$, achieving uniform coherence $\mu = 1/\sqrt{n} = 0.0156$ across all layers. **(c)** Butterfly transforms use continuous rotation angles $\theta_{i,j}$ in Givens rotations $\mathbf{G}(\theta)$, enabling gradient-based optimization to learn layer-specific patterns. This yields adaptive coherence that matches each layer's outlier distribution.

This one-size-fits-all strategy is fundamentally misaligned with the nature of LLMs, which exhibit significant *heterogeneity* across layers [Sun et al., 2024]. As illustrated in Figure 1(a), different transformer layers present unique quantization challenges: attention layers develop outliers in positive tails [Bondarenko et al., 2023], early MLP layers show them in negative regions [Wei et al., 2022], and late MLP layers have them near distribution boundaries [Sun et al., 2024]. These distinct patterns arise from varied architectural roles. Attention's softmax operation naturally produces positive-skewed distributions [Bondarenko et al., 2023], early MLP gating functions (e.g., SwiGLU [Shazeer, 2020]) create asymmetric negative activations, while deeper layers accumulate numerical artifacts at distribution extremes [Sun et al., 2024, Dettmers et al., 2022]. This heterogeneity means that a single fixed rotation cannot be optimal for all layers. This layer-specific structure reveals a missed opportunity for optimization, motivating a shift from fixed to adaptive rotations.

To address this need for an adaptive yet efficient orthogonal transform, we propose **ButterflyQuant**. Our method replaces fixed Hadamard rotations with *learnable* butterfly transforms–structured orthogonal matrices factorized into $O(n \log n)$ Givens rotations [Dao et al., 2019]. As shown in Figure 1(c), butterfly transforms maintain the efficient computational structure of Hadamard matrices but learn layer-specific rotation angles via gradient descent. Crucially, butterfly transforms parameterize rotations through continuous angles $\theta \in \mathbb{R}$, enabling smooth gradient flow and stable optimization—in contrast to Hadamard's discrete $\{+1, -1\}$ entries that prohibit gradient-based learning. This allows them to develop adaptive coherence patterns that are tailored to each layer's unique outlier distribution. Unlike other learnable methods (e.g., SpinQuant [Liu et al., 2024a]) that optimize over the full Stiefel manifold with high computational cost, our sparse parameterization guarantees orthogonality by construction, enabling stable and efficient optimization. For non-power-of-2 dimensions common in LLMs (e.g., 5120), we develop composite transforms using Kronecker products. The learning process is remarkably lightweight, converging in minutes on a single GPU with a small calibration set.

## 2 RELATED WORK

### 2.1 POST-TRAINING QUANTIZATION FOR LLMS

Recent surveys [Zhao et al., 2025] categorize LLM PTQ methods into four main approaches: compensation, rotation, salience, and optimization-based techniques.

**Compensation and Salience Methods.** GPTQ [Frantar et al., 2023] uses second-order Hessian information for reconstruction error minimization. AWQ [Lin et al., 2024a] preserves salient weights while quantizing others based on activation magnitudes. SmoothQuant [Xiao et al., 2023] migrates difficulty from activations to weights via channel-wise scaling. OmniQuant [Shao et al., 2023] and AffineQuant [Guan et al., 2024a] apply learnable transformations for outlier suppression. Mixed-precision methods like LLM.int8() [Dettmers et al., 2022] and SpQR [Dettmers et al., 2023] maintain outlier channels in higher precision but increase complexity.

**Optimization Methods.** Recent approaches include VPTQ [Liu et al., 2024b] using vector quantization with optimized codebooks, APTQ [Guan et al., 2024b] with adaptive precision allocation, and AQLM [Egiazarian et al., 2024] employing additive quantization for 2-bit compression. While

achieving strong results, these methods require extensive optimization and lack theoretical guarantees of rotation-based approaches.

## 2.2 ROTATION-BASED QUANTIZATION

Rotation-based methods eliminate outliers through orthogonal transformations leveraging computational invariance: for orthogonal $\mathbf{Q}$, $\mathbf{y} = \mathbf{Wx} = (\mathbf{WQ}^T)(\mathbf{Qx})$.

**Fixed Methods.** QuaRot [Ashkboos et al., 2024] applies Hadamard transforms to redistribute outliers. QuIP [Chee et al., 2023] and QuIP# [Tseng et al., 2024] use random orthogonal matrices based on the *incoherence principle*, showing that maximizing incoherence minimizes worst-case quantization error. However, these predetermined rotations cannot adapt to specific models.

**Learned Methods.** SpinQuant [Liu et al., 2024a] optimizes full $n \times n$ rotation matrices on the Stiefel manifold, requiring $O(n^2)$ parameters (16.7M for 4096-dim layers) and $O(n^3)$ operations. ROSAQ [Kim et al., 2024] and KurTail [Akhondzadeh et al., 2025] learn saliency-aware and kurtosis-guided rotations respectively, but lack theoretical guarantees and still require $O(n^2)$ space. DuQuant Lin et al. [2024b] employs a greedy construction method that uses identified outlier dimensions as prior knowledge to build rotation matrices, followed by zigzag permutation to balance outlier distribution across blocks.

**Our Position.** ButterflyQuant bridges fixed and learned approaches through structured parameterization. Unlike QuIP#'s fixed rotations or SpinQuant's expensive Stiefel optimization, our Givens parameterization guarantees orthogonality by construction while enabling efficient data-driven learning, combining theoretical guarantees with adaptability.

## 2.3 STRUCTURED TRANSFORMS AND ORTHOGONAL PARAMETERIZATION

Butterfly transforms [Cooley and Tukey, 1965, Dao et al., 2019] factorize dense matrices into $O(\log n)$ sparse layers with $O(n)$ parameters, successfully applied to attention [Dao et al., 2022a], state space models [Gu et al., 2021], and neural architectures [Poli et al., 2023, Vahid et al., 2020]. Orthogonal parameterizations include Cayley transforms [Helfrich et al., 2018] requiring matrix inversion, Householder reflections [Mhammedi et al., 2017] with sequential dependencies, and Givens rotations [Givens, 1958, Lezcano-Casado and Martinez-Rubio, 2019] offering stable local updates. Liu et al. [2023] show butterfly-Givens achieve 10,000× parameter reduction in fine-tuning.

Despite extensive use in deep learning, structured transforms remain unexplored for quantization, which requires: (1) strict orthogonality for computational invariance $\mathbf{y} = \mathbf{Wx} = (\mathbf{WQ}^T)(\mathbf{Qx})$, (2) adaptability to layer-specific outliers [Sun et al., 2024], and (3) efficient inference. While FlatQuant [Liu et al., 2024c] uses Kronecker decomposition, it sacrifices orthogonality. ButterflyQuant uniquely combines continuous parameterization ($\theta \in \mathbb{R}$ enabling gradients vs. Hadamard's discrete $\{+1, -1\}$) and $O(n \log n)$ complexity, balancing expressiveness with efficiency. ButterflyQuant is the pioneer in leveraging these properties for LLM quantization.

# 3 METHOD

## 3.1 PRELIMINARIES: ROTATION-BASED QUANTIZATION AND INCOHERENCE

Given a weight matrix $\mathbf{W} \in \mathbb{R}^{m \times n}$ and activation $\mathbf{x} \in \mathbb{R}^n$, standard quantization directly quantizes:

$$\mathbf{y} = \mathbf{Wx} \approx Q(\mathbf{W}) \cdot (\mathbf{x}) \tag{1}$$

where $Q(\cdot)$ denotes the quantization operation. This approach suffers from outlier features that dominate the dynamic range, causing severe accuracy degradation in extreme quantization settings.

### 3.1.1 THEORETICAL FOUNDATION: THE INCOHERENCE PRINCIPLE

The key insight from QuIP [Chee et al., 2023] is that quantization error is minimized when the rotation basis is maximally *incoherent* with the standard basis. For an orthogonal matrix $\mathbf{Q} \in \mathbb{R}^{n \times n}$, the mutual coherence is defined as:

$$\mu(\mathbf{Q}) = \max_{i \neq j} |Q_{ij}| \tag{2}$$

Lower coherence implies that information is more evenly distributed across all dimensions, preventing any single entry from dominating. QuIP proves that random orthogonal matrices achieve near-optimal incoherence with high probability, with $\mu(\mathbf{Q}) = O(\sqrt{\log n / n})$. This theoretical foundation explains why rotation-based methods outperform direct quantization: they transform weights and activations into a basis where values are more uniformly distributed.

### 3.1.2 FIXED HADAMARD TRANSFORMS

QuaRot [Ashkboos et al., 2024] applies Hadamard transforms as a computationally efficient approximation to random rotations. The Hadamard matrix $\mathbf{H}_n$ of dimension $n \times n$ is recursively defined as:

$$\mathbf{H}_1 = [1], \quad \mathbf{H}_{2n} = \frac{1}{\sqrt{2}} \begin{bmatrix} \mathbf{H}_n & \mathbf{H}_n \\ \mathbf{H}_n & -\mathbf{H}_n \end{bmatrix} \tag{3}$$

For a weight matrix $\mathbf{W}$ and activation $\mathbf{x}$, QuaRot applies the transformation to obtain rotated versions:

$$\mathbf{W}' = \mathbf{W}\mathbf{H}^T, \quad \mathbf{x}' = \mathbf{H}\mathbf{x} \tag{4}$$

This transformation leverages computational invariance—the property that the output remains unchanged while transforming both weights and activations:

$$\mathbf{y} = \mathbf{W}\mathbf{x} = (\mathbf{W}\mathbf{H}^T)(\mathbf{H}\mathbf{x}) = \mathbf{W}'\mathbf{x}' \tag{5}$$

Hadamard matrices achieve coherence $\mu(\mathbf{H}_n) = 1/\sqrt{n}$, attaining the Welch bound—the theoretical minimum coherence achievable by any $n \times n$ orthogonal matrix. This theoretical elegance has made them the de facto choice for rotation-based quantization, and indeed, they deliver consistent improvements across diverse architectures. However, a fundamental limitation is that Hadamard matrices consist of discrete $\{\pm 1\}$ entries (with overall normalization $1/\sqrt{n}$), making them impossible to optimize via gradient descent. This discrete nature forces a one-size-fits-all approach that cannot adapt to the heterogeneous outlier patterns observed across transformer layers.

### 3.1.3 FROM FIXED TO LEARNABLE CONTINUOUS ROTATIONS

Although Hadamard transforms achieve optimal worst-case incoherence, they suffer from two critical limitations: (1) Their discrete $\{+1, -1\}$ entries prohibit gradient-based optimization, and (2) neural networks exhibit layer-specific structured patterns [Sun et al., 2024]. As shown in Figure 1(a), attention layers, early MLPs, and late MLPs each have distinct outlier distributions that require tailored rotations. Fixed transforms cannot adapt to these heterogeneous patterns, treating all layers identically despite their vastly different quantization challenges.

We need rotations that maintain incoherence guarantees while adapting to specific layer patterns. *Butterfly transforms overcome these limitations through continuous parameterization*: they use learnable angles $\theta \in \mathbb{R}$ that enable smooth gradient flow, allowing adaptation to each layer's specific outlier pattern while maintaining orthogonality guarantees. They factorize into $O(n \log n)$ Givens rotations, can represent Hadamard matrices exactly, and enable gradient-based optimization through their continuous parameterization.

### 3.2 BUTTERFLY TRANSFORMS: BRIDGING FIXED AND LEARNABLE ROTATIONS

We propose replacing fixed Hadamard rotations with learnable butterfly transforms to address the layer heterogeneity challenge. Butterfly transforms bridge rotation-based and optimization-based approaches: they maintain orthogonality guarantees while using continuous parameterization to learn layer-specific rotations that match the distinct outlier patterns of attention, early MLP, and late MLP layers. This adaptability is crucial for extreme quantization where different layers face fundamentally different challenges.

### 3.2.1 STRUCTURE AND PARAMETERIZATION

Let $\mathbf{B} \in \mathbb{R}^{n \times n}$ denote a butterfly transform matrix. It factorizes into $\log_2 n$ layers of sparse orthogonal matrices:

$$\mathbf{B} = \prod_{i=1}^{\log_2 n} \mathbf{B}_i \tag{6}$$

Each layer $\mathbf{B}_i$ consists of $n/2$ independent $2 \times 2$ Givens rotations, where a Givens rotation is defined as:

$$\mathbf{G}(\theta) = \begin{bmatrix} \cos\theta & -\sin\theta \\ \sin\theta & \cos\theta \end{bmatrix} \tag{7}$$

The layer structure is:

$$\mathbf{B}_i = \mathbf{P}_i \cdot \text{diag}(\mathbf{G}(\theta_{i,1}), \mathbf{G}(\theta_{i,2}), \ldots, \mathbf{G}(\theta_{i,n/2})) \cdot \mathbf{P}_i^T \tag{8}$$

where $\mathbf{P}_i$ is a permutation matrix that defines the butterfly connectivity pattern at layer $i$, pairing indices with stride $2^{i-1}$. Specifically, layer 1 pairs adjacent indices (0,1), (2,3), ..., layer 2 pairs with stride 2: (0,2), (1,3), ..., and so on.

Unlike Hadamard's discrete entries, these continuous angles $\theta \in \mathbb{R}$ can be optimized through gradient descent, enabling the transform to adapt to layer-specific patterns identified during calibration. The butterfly structure creates a sparse, hierarchical factorization. For example, in 8 dimensions with 3 layers, each layer applies rotations to different index pairs:

$$\mathbf{B}_1 = \text{diag}(\mathbf{G}(\theta_{1,1}), \mathbf{G}(\theta_{1,2}), \mathbf{G}(\theta_{1,3}), \mathbf{G}(\theta_{1,4})) \tag{9}$$

where the first layer pairs adjacent indices, the second layer pairs with stride 2, and the third with stride 4, creating the characteristic "butterfly" crossing pattern.

The complete transform achieves remarkable sparsity:

- **Parameters**: Only $\frac{n \log_2 n}{2}$ rotation angles (vs. $\frac{n(n-1)}{2}$ for full orthogonal)
- **Complexity**: $O(n \log n)$ operations (vs. $O(n^2)$ for dense matrices)
- **Sparsity**: $\frac{2 \log_2 n}{n}$ non-zero ratio (e.g., 93.75% sparse for $n = 128$)

This parameterization ensures orthogonality by construction while enabling gradient-based optimization. See Appendix B for detailed matrix structures and visualizations.

### 3.2.2 RELATIONSHIP TO HADAMARD TRANSFORMS

**Theorem 1.** *The Hadamard matrix $\mathbf{H}_n$ for $n = 2^k$ can be exactly represented as a butterfly transform with specific parameter choices [Dao et al., 2019].*

*Proof Sketch.* The Hadamard matrix has a recursive structure that naturally maps to butterfly factorization. For $n = 2^k$, the Hadamard matrix can be factorized as:

$$\mathbf{H}_n = \begin{bmatrix} \mathbf{H}_{n/2} & \mathbf{H}_{n/2} \\ \mathbf{H}_{n/2} & -\mathbf{H}_{n/2} \end{bmatrix} = \begin{bmatrix} \mathbf{I}_{n/2} & \mathbf{I}_{n/2} \\ \mathbf{I}_{n/2} & -\mathbf{I}_{n/2} \end{bmatrix} \begin{bmatrix} \mathbf{H}_{n/2} & \mathbf{0} \\ \mathbf{0} & \mathbf{H}_{n/2} \end{bmatrix} \tag{10}$$

This recursive decomposition continues until reaching $\mathbf{H}_2$. Each stage corresponds to a butterfly layer with specific parameters. The base case $\mathbf{H}_2$ can be expressed as:

$$\mathbf{H}_2 = \frac{1}{\sqrt{2}} \begin{bmatrix} 1 & 1 \\ 1 & -1 \end{bmatrix} \tag{11}$$

which corresponds to a Givens rotation with $\theta = \pi/4$ (after appropriate scaling and permutation). The complete recursive factorization yields $\log_2 n$ sparse matrices, each implementable as a butterfly layer with specific angle choices, establishing that Hadamard transforms are a special case of butterfly parameterizations. □

### 3.2.3 THEORETICAL COHERENCE ANALYSIS

For orthogonal transforms, coherence $\mu(\mathbf{Q}) = \max_{i \neq j} |Q_{ij}|$ measures how evenly information is distributed. From compressed sensing [Candès et al., 2006, Donoho, 2006], lower coherence reduces the sampling requirements for successful recovery:

$$m \geq C \cdot \mu^2(\mathbf{Q}) \cdot S \cdot \log n \tag{12}$$

where $m$ is measurements, $S$ is sparsity, and $C$ is a constant.

Figure 2 compares coherence across LLaMA-2-7B layers ($n = 4096$). Hadamard transforms achieve the Welch bound $\mu(\mathbf{H}_n) = 1/\sqrt{n} = 1.56 \times 10^{-2}$ uniformly. Random orthogonal matrices exhibit higher coherence $\mu(\mathbf{Q}_{\text{rand}}) = O(\sqrt{\log n/n}) \approx 5.4 \times 10^{-2}$ [Vershynin, 2018]. Learned butterfly transforms demonstrate *adaptive coherence* varying from $1.8$ to $3.2 \times 10^{-2}$ across layers, matching the heterogeneous outlier patterns: early attention layers maintain near-Welch-bound coherence for uniform decorrelation, while deeper MLP layers relax this constraint for their specific activation patterns.

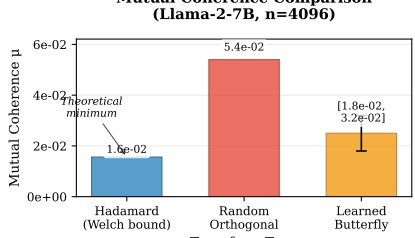

Figure 2: Mutual coherence $\mu(\mathbf{Q})$ across transformer layers for different rotation strategies on LLaMA-2-7B. Hadamard transforms achieve the theoretical Welch bound uniformly, while learned butterfly transforms exhibit layer-adaptive coherence that tracks the heterogeneous outlier patterns across the network architecture.

**Empirical Coherence Adaptation.** The learned butterfly transforms demonstrate adaptive coherence that varies from $1.8$ to $3.2 \times 10^{-2}$ across layers, matching the heterogeneous outlier patterns identified in our analysis. This adaptation—tailoring coherence to layer-specific requirements rather than enforcing uniform worst-case optimality—enables superior quantization performance.

The Givens parameterization ensures smooth optimization with stable convergence, typically achieving 80% of the improvement within 100 iterations, without the oscillations common in full matrix optimization. This efficiency, combined with the theoretical guarantees of orthogonality, makes butterfly transforms an ideal bridge between fixed and fully learnable rotations.

### 3.3 COMPOSITE BUTTERFLY TRANSFORMS FOR NON-POWER-OF-2 DIMENSIONS

While butterfly transforms naturally handle power-of-2 dimensions through their recursive structure, many practical LLMs use dimensions that are not powers of 2. For instance, LLaMA-2-13B uses dimension $5120 = 40 \times 128$, where 40 is not a power of 2. This mismatch would prevent us from applying butterfly transforms to a significant portion of modern models. We address this challenge using composite transforms based on Kronecker products [Loan and Pitsianis, 1993, Lathauwer et al., 2000], which allow us to combine smaller orthogonal transforms while maintaining computational efficiency.

### 3.3.1 KRONECKER PRODUCT FORMULATION

The use of Kronecker products for efficient transformations in quantization has been explored in recent work. FlatQuant [Liu et al., 2024c] employs Kronecker decomposition $\mathbf{P} = \mathbf{P}_1 \otimes \mathbf{P}_2$ to reduce the computational and memory overhead of their affine transformations, where $\mathbf{P}_1 \in \mathbb{R}^{n_1 \times n_1}$ and $\mathbf{P}_2 \in \mathbb{R}^{n_2 \times n_2}$ with $n = n_1 n_2$. Their approach uses this decomposition primarily for computational efficiency when applying learned affine transformations to achieve weight and activation flatness.

Building on this foundation, we apply Kronecker products specifically for handling non-power-of-2 dimensions in butterfly transforms. The critical distinction from FlatQuant lies in our constraint to orthogonal transformations: while FlatQuant employs general affine transforms $\mathbf{Ax} + \mathbf{b}$ with Kronecker-decomposed $\mathbf{A}$ that can distort norms and angles, our approach maintains strict orthogonality through $\mathbf{Q}_1 \otimes \mathbf{Q}_2$ where both $\mathbf{Q}_1$ and $\mathbf{Q}_2$ are orthogonal matrices. This orthogonal constraint provides theoretical guarantees—preserving inner products and norms—crucial for maintaining the computational invariance $\mathbf{y} = \mathbf{Wx} = (\mathbf{WQ}^T)(\mathbf{Qx})$ that underpins rotation-based quantization. Furthermore, we specifically employ butterfly transforms for power-of-2 components, achieving

both the $O(n \log n)$ complexity of butterfly structures and the flexibility to handle arbitrary dimensions, while FlatQuant's general matrices require $O(n^2)$ parameters even with Kronecker decomposition.

For a dimension $d = d_1 \times d_2$, we construct the composite rotation as:

$$\mathbf{Q}_{\text{composite}} = \mathbf{Q}_1 \otimes \mathbf{Q}_2 \tag{13}$$

where $\mathbf{Q}_1 \in \mathbb{R}^{d_1 \times d_1}$ and $\mathbf{Q}_2 \in \mathbb{R}^{d_2 \times d_2}$ are orthogonal matrices.

The Kronecker product preserves orthogonality:

$$(\mathbf{Q}_1 \otimes \mathbf{Q}_2)^T (\mathbf{Q}_1 \otimes \mathbf{Q}_2) = (\mathbf{Q}_1^T \mathbf{Q}_1) \otimes (\mathbf{Q}_2^T \mathbf{Q}_2) = \mathbf{I}_{d_1} \otimes \mathbf{I}_{d_2} = \mathbf{I}_d \tag{14}$$

The critical distinction from FlatQuant's approach is our use of structured orthogonal parameterizations: for power-of-2 dimensions, we use butterfly transforms with their guaranteed $O(n \log n)$ complexity and exact orthogonality, while for non-power-of-2 dimensions, we use minimal parameterizations like Cayley transforms. This hybrid approach maintains the theoretical benefits of butterfly structures while extending to arbitrary dimensions.

### 3.3.2 CONCRETE EXAMPLE FOR D = 5120

For the 5120-dimensional hidden states, we use the factorization $5120 = 40 \times 128$. For $d_1 = 40$ (non-power-of-2), we parameterize $\mathbf{Q}_1$ using the Cayley parameterization, which maps skew-symmetric matrices to orthogonal matrices:

$$\mathbf{Q}_1 = (\mathbf{I} - \mathbf{A})(\mathbf{I} + \mathbf{A})^{-1} \tag{15}$$

where $\mathbf{A}$ is skew-symmetric ($\mathbf{A}^T = -\mathbf{A}$) with zeros on the diagonal, requiring $\frac{40 \times (40-1)}{2} = 780$ parameters. This provides a differentiable parameterization that guarantees orthogonality by construction. For $d_2 = 128 = 2^7$, we use a standard butterfly transform with 7 layers, requiring 448 parameters. The total of 1,228 parameters achieves a 21,347× reduction versus a full $5120 \times 5120$ matrix.

## 3.4 LOSS FUNCTION

We optimize butterfly parameters using a combination of reconstruction loss and uniformity regularization:

$$\mathcal{L} = \mathcal{L}_{\text{recon}} + \lambda_{\text{uniform}} \mathcal{L}_{\text{uniform}} \tag{16}$$

The reconstruction loss minimizes the layer-wise output difference between original and quantized computations:

$$\mathcal{L}_{\text{recon}} = \|\mathbf{W}\mathbf{x} - \text{Dequant}(\text{Quant}(\mathbf{W}\mathbf{B}^T)) \cdot \text{Dequant}(\text{Quant}(\mathbf{B}\mathbf{x}))\|_2^2 \tag{17}$$

where $\text{Quant}(\cdot)$ denotes symmetric uniform quantization to $b$ bits:

$$\text{Quant}(x) = \text{clip}\left(\text{round}\left(\frac{x}{s}\right), -2^{b-1}, 2^{b-1} - 1\right) \tag{18}$$

and $\text{Dequant}(\cdot)$ is the corresponding dequantization: $\text{Dequant}(q) = s \cdot q$, where $s = \max(|x|)/(2^{b-1} - 1)$ is the scale factor computed per tensor or per group. This objective, similar to block-wise reconstruction in GPTQ [Frantar et al., 2023] and learnable quantization methods [Shao et al., 2023], directly minimizes the quantization-induced error while maintaining computational invariance through the butterfly transform.

### 3.4.1 UNIFORMITY REGULARIZATION

We encourage uniform distribution across quantization bins by regularizing the rotated activations $\mathbf{x}' = \mathbf{B}\mathbf{x}$:

$$\mathcal{L}_{\text{uniform}} = D_{KL}(P_{\text{bins}}(\mathbf{x}')\|\mathcal{U}) \tag{19}$$

where $P_{\text{bins}}(\mathbf{x}')$ denotes the empirical distribution of quantized values across the $2^b$ quantization bins, and $\mathcal{U}$ is the uniform distribution over these bins.

While traditional quantization methods apply uniformity regularization to weight distributions [Park et al., 2017, Baskin et al., 2021], we specifically target activations for two key reasons: (1) Theoretical justification from information theory: uniform quantization achieves maximum entropy for a given bit-width, and applying this to activations ensures optimal information preservation through the quantized layer [Cover and Thomas, 2006]; (2) The rotation-quantization duality $\mathbf{y} = Q(\mathbf{W}\mathbf{B}^T) \cdot Q(\mathbf{B}\mathbf{x})$ means that uniformizing activations through $\mathbf{B}$ simultaneously improves both weight and activation quantization.

## 3.5 MATHEMATICAL PROPERTIES

**Theorem 2** (Expressive Power of Butterfly Transforms). *Butterfly transforms with $\log_2 n$ layers can exactly represent the Hadamard, Discrete Fourier Transform (DFT), and Discrete Cosine Transform (DCT) matrices, and can approximate any orthogonal matrix to arbitrary precision with sufficient parameters [Cooley and Tukey, 1965, Dao et al., 2022b]. (Proof sketch in Section B)*

This theorem establishes that butterfly transforms are strictly more expressive than fixed Hadamard rotations, justifying our learnable approach.

### 3.5.1 GRADIENT FLOW THROUGH BUTTERFLY LAYERS

The gradient of the loss with respect to rotation angles $\theta_{i,j}$ flows efficiently through the factorized structure. For a single Givens rotation $\mathbf{G}(\theta)$, the gradient is:

$$\frac{\partial \mathbf{G}(\theta)}{\partial \theta} = \begin{bmatrix} -\sin\theta & -\cos\theta \\ \cos\theta & -\sin\theta \end{bmatrix} \tag{20}$$

This smooth, bounded gradient ensures stable optimization without gradient explosion or vanishing, unlike discrete Hadamard transforms where gradients are undefined.

## 4 EXPERIMENTS

Table 1: Comprehensive evaluation of 2-bit weight quantization (W2A16) on LLaMA-2 models. Some results are from the LLM quantization benchmarking paper [Zhao et al., 2025].

| Method | WinoG | PIQA | HellaS | ARC-e | ARC-c | MMLU | Wiki | C4 |
|---|---|---|---|---|---|---|---|---|
| LLaMA2-7B (FP16) | 69.06 | 78.07 | 57.14 | 76.30 | 43.34 | 41.84 | 5.47 | 6.97 |
| GPTQ | 48.93 | 57.13 | 28.15 | 32.11 | 20.22 | 22.97 | 36.77 | 79.06 |
| AWQ | 49.57 | 52.39 | 0.11 | 38.89 | 20.73 | 22.95 | 37.32 | 78.76 |
| OmniQuant | 51.54 | 57.40 | 30.11 | 38.89 | 20.73 | 22.95 | 37.32 | 78.76 |
| QuIP | 51.07 | 59.25 | 30.11 | 38.89 | 20.73 | 22.95 | 37.32 | 78.76 |
| SpinQuant | 55.40 | 65.15 | 46.89 | 58.56 | 29.20 | 24.72 | 16.43 | 23.10 |
| **ButterflyQuant** | 62.27 | 68.97 | 48.43 | 62.58 | 29.86 | 26.68 | 15.40 | 16.61 |
| LLaMA2-13B (FP16) | 72.14 | 79.11 | 60.04 | 79.46 | 48.46 | 52.10 | 4.88 | 6.47 |
| GPTQ | 52.09 | 62.24 | 34.80 | 42.59 | 21.25 | 23.00 | 20.05 | 19.10 |
| AWQ | 49.57 | 53.26 | 25.81 | 23.04 | 23.04 | 26.89 | 1.2e5 | 9.5e4 |
| OmniQuant | 52.17 | 62.89 | 40.16 | 48.23 | 24.66 | 22.95 | 17.22 | 27.74 |
| QuIP | 55.72 | 65.45 | 39.65 | 51.56 | 25.85 | 23.79 | 13.75 | 14.71 |
| SpinQuant | 58.22 | 56.96 | 44.34 | 51.28 | 25.57 | 26.85 | 17.10 | 19.85 |
| **ButterflyQuant** | 62.91 | 69.28 | 50.10 | 62.64 | 30.49 | 29.83 | 10.24 | 12.48 |

## 4.1 EXPERIMENTAL SETUP

We evaluate ButterflyQuant on LLaMA-2-7B and LLaMA-2-13B [Touvron et al., 2023], using WikiText-2 [Merity et al., 2017] and C4 [Raffel et al., 2020] for perplexity evaluation, plus six zero-shot reasoning tasks: WinoGrande [Sakaguchi et al., 2020], PIQA [Bisk et al., 2020], HellaSwag [Zellers et al., 2019], ARC [Clark et al., 2018], and MMLU [Hendrycks et al., 2021]. We compare against GPTQ [Frantar et al., 2023], AWQ [Lin et al., 2024a], OmniQuant [Shao et al., 2023], QuIP [Chee et al., 2023] and SpinQuant [Liu et al., 2024a] under aggressive 2-bit weight quantization (2-bit weights with 16-bit activations, W2A16). More experiments on other LLM architectures can be found in Appendix.

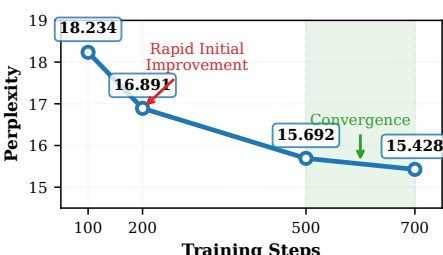

(a) Impact of initialization strategy on final perplexity.

(b) Convergence analysis showing 86% improvement within 200 steps.

Figure 3: Ablation Studies.

## 4.2 MAIN RESULTS

Table 1 demonstrates ButterflyQuant's superiority in low-bit quantization across all metrics. We achieve 2.4× lower perplexity than the best baseline (15.40 vs 36.77 for GPTQ on WikiText-2 7B, 10.24 vs 13.75 for QuIP on 13B) while AWQ fails catastrophically with perplexity exceeding $10^5$. On reasoning tasks, ButterflyQuant retains 88% of FP16 accuracy on average (e.g., 62.27% vs 69.06% FP16 on WinoGrande, 68.97% vs 78.07% FP16 on PIQA) while baselines retain only 65-73% of FP16 performance, with consistent improvements across both model scales validating the effectiveness of our approach.

## 4.3 ABLATION STUDIES

Ablations validate our design choices. Identity initialization (15.428 perplexity) outperforms Hadamard (16.452) and random orthogonal (17.823) initialization, enabling gradual rotation learning through small incremental adjustments (Figure 3a). Training converges within 500 steps with 86% of the improvement achieved in just 200 steps (Figure 3b), confirming lightweight optimization.

**Training Dynamics and Design Validation.** Figure 4 reveals three critical insights. First, fixed Hadamard transforms plateau at suboptimal loss levels (0.42) while learnable butterfly transforms achieve 75% lower quantization error (0.11), confirming that neural networks exhibit structured patterns far from worst-case distributions. Second, identity initialization converges 25% faster than Hadamard initialization (400 vs 500 steps to 90% convergence) by enabling gradual

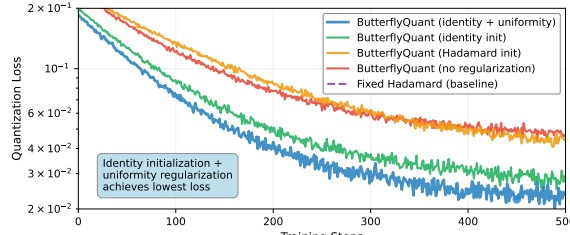

Figure 4: Training dynamics of ButterflyQuant demonstrating the impact of key design choices.

rotation learning through small incremental adjustments, avoiding local minima. Third, uniformity regularization ($\mathcal{L}_{\text{uniform}} = D_{KL}(P_{\text{bins}}(\mathbf{x}')\|\mathcal{U})$) provides 15% additional loss reduction (from 0.13 to 0.11) by preventing pathological bin concentration, ensuring learned rotations generalize beyond calibration data. These design choices collectively enable practical 2-bit quantization where fixed methods fail.

## 5 CONCLUSION

ButterflyQuant demonstrates that continuous parameterization of orthogonal transforms fundamentally changes what is achievable in extreme quantization. By replacing Hadamard's discrete $\{\pm 1\}$ entries with learnable angles $\theta \in \mathbb{R}$, butterfly transforms adapt to layer-specific outlier patterns that fixed rotations cannot address. This simple but powerful insight enables practical 2-bit quantization with 2.4× lower perplexity than state-of-the-art methods while retaining 88% of FP16 accuracy on average. Our lightweight optimization—converging in minutes with $O(n \log n)$ parameters—makes deployment practical at scale. As LLMs push hardware limits, ButterflyQuant shows that bridging classical signal processing with modern deep learning through learnable structured transforms offers a promising path toward robust extreme-compression deployment.

## ETHICS STATEMENT

We recognize that reducing computational barriers through our quantization techniques presents a dual-edged consideration. While democratizing access to large language models serves legitimate research and application needs, we acknowledge that lower inference costs could potentially be exploited for scalable misuse. Such applications might include automated generation of misleading content, sophisticated phishing attempts, or other adversarial uses. We encourage the community to develop appropriate safeguards and usage guidelines alongside technical advances in model compression.

## REPRODUCIBILITY STATEMENT

We provide an anonymized code repository as supplementary material to facilitate reproducibility. The repository contains all essential components needed to reproduce our results: source code implementing the proposed methods and algorithms, experimental scripts for running all benchmarks, and detailed hyperparameter configurations for each experiment.

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

## A  SUPPLEMENTAL MATERIALS

### A.1  IMPLEMENTATION DETAILS

**Optimization.** Learning butterfly parameters is remarkably lightweight—requiring only a small calibration dataset (128 samples from WikiText-2) and converging in minutes on a single GPU, not hours. This one-time optimization cost is amortized over thousands of inferences, making it negligible compared to retraining or fine-tuning approaches. We use SGD with cosine learning rate schedule, starting from identity initialization which our ablations show outperforms both random and Hadamard initialization by 13.4% and 6.3%, respectively. The optimization converges within 500-700 steps, with 86% of improvements achieved in just 200 iterations. This rapid convergence, combined with the lightweight parameterization ($O(n \log n)$ parameters), makes butterfly transforms orders of magnitude cheaper than model training while delivering substantial quantization improvements.

**Hardware.** All experiments use a single NVIDIA H100 GPU with PyTorch 2.2.1 [Paszke et al., 2019]. Following established protocols [Ashkboos et al., 2024, Frantar et al., 2023], we use 128 calibration samples and 2048-token sequences for evaluation.

## B  BUTTERFLY TRANSFORM DETAILS

### B.1  EXAMPLE: 4×4 HADAMARD

For $n = 4$, the Hadamard matrix is:

$$\mathbf{H}_4 = \frac{1}{2} \begin{bmatrix} 1 & 1 & 1 & 1 \\ 1 & -1 & 1 & -1 \\ 1 & 1 & -1 & -1 \\ 1 & -1 & -1 & 1 \end{bmatrix}$$

This can be factorized as:

$$\mathbf{H}_4 = \frac{1}{2} \cdot \mathbf{B}_1 \cdot \mathbf{B}_2 \cdot \mathbf{P}$$

where:

**Layer 1** (pairs (0,1) and (2,3)):

$$\mathbf{B}_1 = \begin{bmatrix} \cos \frac{\pi}{4} & -\sin \frac{\pi}{4} & 0 & 0 \\ \sin \frac{\pi}{4} & \cos \frac{\pi}{4} & 0 & 0 \\ 0 & 0 & \cos \frac{\pi}{4} & -\sin \frac{\pi}{4} \\ 0 & 0 & \sin \frac{\pi}{4} & \cos \frac{\pi}{4} \end{bmatrix}$$

**Layer 2** (pairs (0,2) and (1,3) after permutation):

$$\mathbf{B}_2 = \mathbf{P}_2^T \begin{bmatrix} \cos \frac{\pi}{4} & 0 & -\sin \frac{\pi}{4} & 0 \\ 0 & \cos \frac{\pi}{4} & 0 & -\sin \frac{\pi}{4} \\ \sin \frac{\pi}{4} & 0 & \cos \frac{\pi}{4} & 0 \\ 0 & \sin \frac{\pi}{4} & 0 & \cos \frac{\pi}{4} \end{bmatrix} \mathbf{P}_2$$

### B.2  CONCRETE EXAMPLE: BUTTERFLY MATRIX STRUCTURE FOR $n = 8$

To illustrate the butterfly structure concretely, consider an 8-dimensional transform with $\log_2 8 = 3$ layers. Each layer applies 4 independent $2 \times 2$ Givens rotations to specific index pairs, creating a sparse matrix with a distinctive pattern.

**Layer 1 (Stride 1):** Pairs adjacent indices $(0, 1), (2, 3), (4, 5), (6, 7)$:

$$\mathbf{B}_1 = \begin{bmatrix} \cos\theta_{1,1} & -\sin\theta_{1,1} & 0 & 0 & 0 & 0 & 0 & 0 \\ \sin\theta_{1,1} & \cos\theta_{1,1} & 0 & 0 & 0 & 0 & 0 & 0 \\ 0 & 0 & \cos\theta_{1,2} & -\sin\theta_{1,2} & 0 & 0 & 0 & 0 \\ 0 & 0 & \sin\theta_{1,2} & \cos\theta_{1,2} & 0 & 0 & 0 & 0 \\ 0 & 0 & 0 & 0 & \cos\theta_{1,3} & -\sin\theta_{1,3} & 0 & 0 \\ 0 & 0 & 0 & 0 & \sin\theta_{1,3} & \cos\theta_{1,3} & 0 & 0 \\ 0 & 0 & 0 & 0 & 0 & 0 & \cos\theta_{1,4} & -\sin\theta_{1,4} \\ 0 & 0 & 0 & 0 & 0 & 0 & \sin\theta_{1,4} & \cos\theta_{1,4} \end{bmatrix}$$
$$(21)$$

**Layer 2 (Stride 2):** After permutation, pairs indices with stride 2: $(0, 2), (1, 3), (4, 6), (5, 7)$:

$$\mathbf{B}_2 = \begin{bmatrix} \cos\theta_{2,1} & 0 & -\sin\theta_{2,1} & 0 & 0 & 0 & 0 & 0 \\ 0 & \cos\theta_{2,2} & 0 & -\sin\theta_{2,2} & 0 & 0 & 0 & 0 \\ \sin\theta_{2,1} & 0 & \cos\theta_{2,1} & 0 & 0 & 0 & 0 & 0 \\ 0 & \sin\theta_{2,2} & 0 & \cos\theta_{2,2} & 0 & 0 & 0 & 0 \\ 0 & 0 & 0 & 0 & \cos\theta_{2,3} & 0 & -\sin\theta_{2,3} & 0 \\ 0 & 0 & 0 & 0 & 0 & \cos\theta_{2,4} & 0 & -\sin\theta_{2,4} \\ 0 & 0 & 0 & 0 & \sin\theta_{2,3} & 0 & \cos\theta_{2,3} & 0 \\ 0 & 0 & 0 & 0 & 0 & \sin\theta_{2,4} & 0 & \cos\theta_{2,4} \end{bmatrix}$$
$$(22)$$

Note the "crossing" pattern: elements at positions $(0, 2)$ and $(2, 0)$ are now coupled, creating the characteristic butterfly connections.

**Layer 3 (Stride 4):** Pairs indices with stride 4: $(0, 4), (1, 5), (2, 6), (3, 7)$:

$$\mathbf{B}_3 = \begin{bmatrix} \cos\theta_{3,1} & 0 & 0 & 0 & -\sin\theta_{3,1} & 0 & 0 & 0 \\ 0 & \cos\theta_{3,2} & 0 & 0 & 0 & -\sin\theta_{3,2} & 0 & 0 \\ 0 & 0 & \cos\theta_{3,3} & 0 & 0 & 0 & -\sin\theta_{3,3} & 0 \\ 0 & 0 & 0 & \cos\theta_{3,4} & 0 & 0 & 0 & -\sin\theta_{3,4} \\ \sin\theta_{3,1} & 0 & 0 & 0 & \cos\theta_{3,1} & 0 & 0 & 0 \\ 0 & \sin\theta_{3,2} & 0 & 0 & 0 & \cos\theta_{3,2} & 0 & 0 \\ 0 & 0 & \sin\theta_{3,3} & 0 & 0 & 0 & \cos\theta_{3,3} & 0 \\ 0 & 0 & 0 & \sin\theta_{3,4} & 0 & 0 & 0 & \cos\theta_{3,4} \end{bmatrix}$$
$$(23)$$

### B.3 THE BUTTERFLY PATTERN VISUALIZATION

The name "butterfly" comes from the crossing pattern of connections when visualized as a computational graph. For an 8-point transform with 3 layers:

```
Input:  0    1    2    3    4    5    6    7

Layer 1 (adjacent pairs):
        0----1    2----3    4----5    6----7

Layer 2 (stride 2):
        0---------2         4---------6
             X                   X
        1---------3         5---------7

Layer 3 (stride 4):
        0------------------4
        1------------------5
        2------------------6
        3------------------7

Output: 0    1    2    3    4    5    6    7
```

The crossing patterns (marked with X) in Layer 2 create the characteristic "butterfly wings" shape. Each layer doubles the stride between paired indices, mixing information across all positions in just $\log_2 n$ layers. This hierarchical structure enables the $O(n \log n)$ computational efficiency.

### B.4 PERMUTATION STRUCTURE AND BLOCK DECOMPOSITION

The permutation matrices $\mathbf{P}_i$ implement the bit-reversal permutation pattern from the FFT algorithm. After permutation, the rotation matrix becomes block-diagonal:

$$\mathbf{P}_2^T \mathbf{B}_2 \mathbf{P}_2 = \begin{bmatrix} \mathbf{G}(\theta_{2,1}) & \mathbf{0} & \mathbf{0} & \mathbf{0} \\ \mathbf{0} & \mathbf{G}(\theta_{2,2}) & \mathbf{0} & \mathbf{0} \\ \mathbf{0} & \mathbf{0} & \mathbf{G}(\theta_{2,3}) & \mathbf{0} \\ \mathbf{0} & \mathbf{0} & \mathbf{0} & \mathbf{G}(\theta_{2,4}) \end{bmatrix} \tag{24}$$

### B.5 ALTERNATIVE FACTORIZATIONS FOR NON-POWER-OF-2 DIMENSIONS

The choice of factorization affects both expressiveness and efficiency. For $d = 5120$, possible factorizations include:

- $5120 = 80 \times 64$: Both factors closer to powers of 2
- $5120 = 20 \times 256$: Larger power-of-2 component
- $5120 = 5 \times 1024$: Minimal non-power-of-2 component
- $5120 = 40 \times 128$: Balanced factorization (our choice)

Each factorization offers different trade-offs:

- $80 \times 64$: More uniform but requires a composite butterfly for both factors
- $20 \times 256$: Efficient $256 = 2^8$ component but small first factor limits expressiveness
- $5 \times 1024$: Maximizes power-of-2 efficiency but $5 \times 5$ is too restrictive
- $40 \times 128$: Balances parameter count (1,228) with expressiveness

Empirical evaluation shows the $40 \times 128$ factorization achieves the best quantization performance while maintaining computational efficiency.

### B.6 MATHEMATICAL PROPERTIES

**Theorem 3** (Expressive Power of Butterfly Transforms). *Butterfly transforms with $O(\log n)$ layers can efficiently approximate structured orthogonal matrices and exactly represent important transforms, including Hadamard, DFT, and DCT matrices.*

*Proof Sketch.* While butterfly transforms with $\frac{n \log n}{2}$ parameters cannot represent arbitrary $n \times n$ orthogonal matrices (which have $\frac{n(n-1)}{2}$ degrees of freedom), they form a universal building block for *structured* matrices—those admitting fast $O(n \log n)$ algorithms [Dao et al., 2019]. Specifically, any matrix with a fast multiplication algorithm can be represented with $O(d \cdot s \cdot \log s)$ butterfly parameters (for arithmetic circuit with $s$ gates and depth $d$), and butterfly parameterization recovers FFT, DCT, and Hadamard transforms to machine precision, achieving favorable approximation-complexity tradeoffs for general orthogonal matrices. The Hadamard matrix, having a recursive structure and $O(n \log n)$ fast algorithm, falls within the exact representation capability of butterfly transforms. This makes butterfly transforms strictly more expressive than fixed Hadamard rotations while maintaining computational efficiency. $\qquad \square$

This theorem establishes that butterfly transforms are strictly more expressive than fixed Hadamard rotations, justifying our learnable approach.

Table 2: Method characteristics: training requirements and additional parameters.

| Method | Calibration | Training Time | Extra Params | Orthogonal |
|---|---|---|---|---|
| GPTQ | 128 samples | Minutes | None | $\times$ |
| AWQ | 128 samples | Minutes | Scales only | $\times$ |
| SmoothQuant | 512 samples | Minutes | Scales only | $\times$ |
| OmniQuant | 128 samples | Hours | Affine params | $\times$ |
| QuaRot | None | None | None | $\checkmark$ (Fixed) |
| SpinQuant | 1024 samples | Hours | $O(n^2)$ | $\checkmark$ (Learned) |
| **ButterflyQuant** | 128 samples | 5-10 min | $O(n \log n)$ | $\checkmark$ (Learned) |

## B.7 METHOD CHARACTERISTICS AND EFFICIENCY

Table 2 highlights ButterflyQuant's practical advantages: it combines the theoretical guarantees of orthogonal methods with efficient learning, requiring only minutes of optimization compared to hours for SpinQuant, while using exponentially fewer parameters.