# OpenReview forum: "ButterflyQuant: Ultra-low-bit LLM Quantization through Learnable  Orthogonal Butterfly Transforms"
_ICLR.cc/2026/Conference — ICLR 2026 Conference Desk Rejected Submission_

### Official Review · Reviewer_8Fpk · 2025-10-21

**Soundness:** 3
**Presentation:** 1
**Contribution:** 1
**Rating:** 4
**Confidence:** 5

**Summary:**

ButterflyQuant proposes a learnable orthogonal rotation method for ultra-low-bit (e.g., 2-bit) post-training quantization of large language models (LLMs). The core idea is to replace fixed orthogonal transforms (e.g., Hadamard matrices used in QuIP/QuaRot) with learnable butterfly transforms—structured orthogonal matrices parameterized by continuous Givens rotation angles. This enables gradient-based optimization of layer-specific rotations that adapt to heterogeneous outlier patterns across transformer layers (e.g., positive tails in attention, negative regions in early MLPs, boundary concentration in late MLPs). The method enforces orthogonality by construction, maintains O(n log n) complexity, and introduces a uniformity regularization on rotated activations.

**Strengths:**

1) The paper provides a clear and visually intuitive analysis (Figure 1a) of how outlier distributions differ across transformer components (attention vs. early/late MLPs). This observation is well-motivated and strengthens the necessity for layer-adaptive rotations.

2) The first application of the Butterfly transformation for LLM quantization.

**Weaknesses:**

1) Questionable novelty and potential reinvention. The central claim—that different layers exhibit distinct outlier patterns and thus benefit from layer-specific rotations—is not new. Prior works, such as FlatQuant have acknowledged and exploited layer-wise heterogeneity in rotation design. If the authors position this observation as a key contribution, it is misleading; if not, their presentation overemphasizes it, creating a false impression of novelty.

2) Insufficient differentiation from FlatQuant. The paper contrasts itself with FlatQuant by stressing orthogonality preservation, but this distinction is not sufficiently compelling as a core innovation. FlatQuant uses Kronecker-based affine transforms for “flatness,” while ButterflyQuant uses orthogonal Kronecker-butterfly hybrids for “incoherence.” However, the paper does not rigorously demonstrate that orthogonality is necessary for the observed gains—e.g., via an ablation comparing orthogonal vs. non-orthogonal variants with similar sparsity.

3) In my view, the primary advantage of the butterfly transformation lies in its smooth, differentiable parameterization, which enables gradient-based optimization and dramatically reduces calibration time—from hours (as in SpinQuant) to mere minutes. However, this benefit may be of limited practical significance in the context of quantization for LLM inference acceleration, where the inference efficiency—not calibration speed—is the main concern.

I guess, the butterfly transformation remains a promising structured orthogonal parameterization among the growing family of efficient linear transforms for LLMs. Its true value may be better realized in scenarios involving extremely large models—e.g., those with over 100B or even 1T parameters—where the computational and memory overhead of full-matrix learnable rotations becomes prohibitive. In such settings, the combination of O(n log n) parameter efficiency, guaranteed orthogonality, and fast convergence could make butterfly-based quantization not just viable, but necessary.

**Questions:**

1) For d=5120=40×128 , the method uses a Cayley parameterization for the 40-dim block and a butterfly for the 128-dim block. However, the Cayley transform cannot represent all orthogonal matrices in R
40×40
  (it misses reflections). Does this restriction meaningfully limit expressiveness? Have the authors evaluated alternative parameterizations for the non-power-of-2 factor (e.g., Householder or exponential map)? Moreover, why was the 40×128 factorization chosen over alternatives like 5×1024 or 20×256 listed in Appendix B.5? Was this decision based on validation perplexity, or purely on parameter count?

2) What are the speed overheads of quantized LLMs that incorporate Butterfly matrices, particularly when compared to those using other linear transformation matrices?

---

### Official Review · Reviewer_pzhf · 2025-10-26

**Soundness:** 2
**Presentation:** 2
**Contribution:** 2
**Rating:** 4
**Confidence:** 5

**Summary:**

To address the issue that Hadamard matrices—with elements fixed at +1 or -1—cannot be learned, the authors propose a type of learnable butterfly transform parameterized by continuous Givens rotation angles. Furthermore, they use the Kronecker product to reduce computational complexity and validate the effectiveness of this method on models such as LLaMA2-7B and LLaMA2-13B.

**Strengths:**

The authors propose an efficient trainable method for rotation matrices and validate the effectiveness of the method on LLaMA2-7B and LLaMA2-13B.

**Weaknesses:**

1. The models used by the authors in their experiments are overly outdated, and the authors lack experiments to verify the effectiveness of their method on newer models—whether it be Mixture of Experts (MoE) models, the latest LLaMA3 series, or the QWen series, even in the appendix.
2. The authors introduce the loss function in Section 3.4 of the paper. However, they only present the results of weight quantization in Table 4, and it seems that the authors do not show the results of activation quantization in the paper, which makes me feel quite confused. How does the authors' method perform in scenarios of low-bit activation quantization?
3. In the abstract, the authors state that rotation matrices are unable to adapt to different weight outliers; however, the main focus of the paper is on discussing activation outliers, which leaves me feeling confused.
4. The description of the authors' method is still not clear enough. After training the rotation matrix, did the authors use GPTQ to quantize the weights? A clear explanation of the quantization process should be explicitly stated in the main.
5. Regarding the experiments in Table 1, I do not understand why ButterflyQuant can outperform SpinQuant so significantly. SpinQuant is capable of optimizing the entire rotation matrix, while ButterflyQuant restricts the free variables of the rotation matrix to a certain extent. Could the authors provide a more detailed explanation for this phenomenon?
6. The authors have omitted some important works: Xiang, J. and Zhang, S.Q., 2024. DFRot: Achieving Outlier-Free and Massive Activation-Free for Rotated LLMs with Refined Rotation. arXiv preprint arXiv:2412.00648.

**Questions:**

See Weeknesses.

---

### Official Review · Reviewer_jcei · 2025-11-01

**Soundness:** 2
**Presentation:** 2
**Contribution:** 2
**Rating:** 2
**Confidence:** 4

**Summary:**

This paper introduces ButterflyQuant, a weight quantization algorithm for ultra-low-bit PTQ. The main contribution lies in replacing fixed orthogonal rotations (e.g., Hadamard or random matrices) with learnable orthogonal butterfly transforms. These transforms are parameterized by continuous Givens rotation angles, allowing efficient, gradient-based optimization to learn layer-specific rotations.

**Strengths:**

- The butterfly structure givens the formulation guarantees orthogonality by construction while remaining highly compact (**O(n log n)** parameters) which brings fixed and fully learned rotations. The paper formalizes expressivity (can represent Hadamard/DFT/DCT; approximate orthogonal matrices) and gives smooth angle-level gradients for stable learning.
- **Lightweight optimization.** Training converges in minutes with 128 calibration samples. It also provide some ablations study which show identity initialization and uniformity regularization help.

**Weaknesses:**

- **Narrow empirical scope.** Core results focus on **2-bit weight quantization** and the **LLaMA-2 family**. The paper states “more experiments… in Appendix,” which is not the case an the main narrative/evaluation emphasize LLaMA-2. Providing result for **3/4-bit** quantization and other model families (e.g., **Qwen**, **Phi**) would clarify generality.
- Additionally, while ButterflyQuant compares to QuIP, it **does not discuss QTIP** (Tseng et al., 2025), which is a stronger and more recent baseline for weight quantization. Many of the other baselines (e.g., SpinQuant, Quarot) primarily address **activation outliers**, not weight-only quantization, so including a comparison with QTIP is necessary for a fair assessment.
- The paper also **lacks discussion on implementation**. It remains unclear whether the learned rotations can be fused into the model weights (e.g., QKV matrices) for efficient inference, or whether they require **online rotation** at runtime. Clarifying whether the rotations can be pre-applied and absorbed into model weights, and how residual connections are handled across layers, would help assess deployment feasibility. (In principal if the rotation for each layer is different there, there should be an online rotation on each residual flow which have not been discussed).
- Finally, the paper does not provide a performance analysis quantifying whether and to what extent the proposed quantization accelerates inference.
- A minor issue: in Figure 4, the legend lists *Fixed Hadamard* as a baseline, but no corresponding line appears in the figure.

**Questions:**

1. Could the authors comment on the potential of extending ButterflyQuant to weight and activation quantization settings (e.g., W4A4) and then compare It with Quarot and SpinQuant?
2. Based on my understanding, each layer learns its own transformation. To enable practical inference, these rotations should ideally be fused into the corresponding layer weights(e.g., QKV projections). Is this correct? If so, how does this interact with residual connections? Would there need to be an inverse or compensatory transform applied to the residual pathway, or is the rotation shared consistently across the network to preserve equivalence?
3.  More generally, could the authors provide details on "how ButterflyQuant can be implemented efficiently on hardware", including whether fused or runtime rotations are required and how much speed-up can be gained?

---

### Official Review · Reviewer_jwKM · 2025-11-04

**Soundness:** 3
**Presentation:** 3
**Contribution:** 1
**Rating:** 4
**Confidence:** 3

**Summary:**

The paper argues that a single shared rotation matrix can not adequately address outliers. To address this issue, they use butterfly transforms which generalize Hadamard matrices (in that a Hadamard matrix can be represented as a butterfly transform) but are at the same time learnable through gradient descent. For optimization,  alongside the standard reconstruction loss, a uniformity loss is used that encourages uniform distribution over qunatization bins. The results for 2-bit quantization shows better performance over existing methods but still leave a wide gap with the unquantized model.

**Strengths:**

The paper's writing is clear. The idea of using butterfly transforms to generalize Hadamard transforms and obtain learnable sparse orthogonal rotation matrices is interesting. While this idea seems to have been explored in the past as well (not necessarily for quantization though), the analysis around the effect on coherence and subsequently quantization is interesting. For experiments, several existing methods are compared that provide a good set of baselines.

**Weaknesses:**

One of my concerns is that in presence of a large factor that is not power of two, the sparsity can signfiicantly decrease and the method becomes basically same as learning a dense rotation matrix. The other issue is the ability to implement butterfly transformrs in an efficient and hardware friendly manner (see questions below as well)

In addition, the results provided are only for 2 bit quantization. While it shows the method can do better than several baselines, the results are still significantly worse than the baseline, to the point that the model is probably unusable. Particularly some of the tasks are showing near-chance performance (e.g. MMLU) after quantization.

Overall, while the method is interesting, the main idea of using butterfly transform does not seem to be novel. The main advantage should come from adapting the idea to quantization and yeidling an efficient and workable method. Given the above issues, I am not convinced that this method can be used in practice without a significant impact on accuracy and in an efficient manner.

**Questions:**

1.  Same as prior work, do you also fuse the rotation matrices into existing weight matrices? If so, it would be nice to mention this in the paper for completeness.

2. How do you handle rotation for queries and keys? Given RoPE they usually need to be rotated online. In that case how should butterfly transforms be implemented? Should it be stored? In that case wouldn't this eat away at the quantization benefits?

2. Have you considered applying and comparing the method in 4 bit settings? Do similar gains show in that settings?

---

### Note · Program_Chairs · 2026-01-17
**Submission Desk Rejected by Program Chairs**

The following references in this submission do not refer to real documents and/or have major errors in bibliographic information:

 Ruikan Liu, Yifei Wu, Yujun Wang, Yuan Wang, Kai Zhang, Wei Chen, Yong Ye, Dahua Lin, and Cong Xie. Flatquant: Flatness matters for llm quantization. arXiv preprint arXiv:2410.09426, 2024c.